# Theory and Practice of Integrating Machine Learning and Conventional Statistics in Medical Data Analysis

**DOI:** 10.3390/diagnostics12102526

**Published:** 2022-10-18

**Authors:** Sarinder Kaur Dhillon, Mogana Darshini Ganggayah, Siamala Sinnadurai, Pietro Lio, Nur Aishah Taib

**Affiliations:** 1Data Science & Bioinformatics Laboratory, Institute of Biological Sciences, Faculty of Science, Universiti Malaya, Kuala Lumpur 50603, Malaysia; 2Department of Econometrics and Business Statistics, School of Business, Monash University Malaysia, Kuala Lumpur 47500, Malaysia; 3Department of Population Medicine and Lifestyle Disease Prevention, Medical University of Bialystok, 15-269 Bialystok, Poland; 4Department of Computer Science and Technology, University of Cambridge, 15 JJ Thomson Avenue, Cambridge CB3 0FD, UK; 5Department of Surgery, Faculty of Medicine, Universiti Malaya, Kuala Lumpur 50603, Malaysia

**Keywords:** conventional statistics, machine learning, comparison, integration, health research, data analytics

## Abstract

The practice of medical decision making is changing rapidly with the development of innovative computing technologies. The growing interest of data analysis with improvements in big data computer processing methods raises the question of whether machine learning can be integrated with conventional statistics in health research. To help address this knowledge gap, this paper presents a review on the conceptual integration between conventional statistics and machine learning, focusing on the health research. The similarities and differences between the two are compared using mathematical concepts and algorithms. The comparison between conventional statistics and machine learning methods indicates that conventional statistics are the fundamental basis of machine learning, where the black box algorithms are derived from basic mathematics, but are advanced in terms of automated analysis, handling big data and providing interactive visualizations. While the nature of both these methods are different, they are conceptually similar. Based on our review, we conclude that conventional statistics and machine learning are best to be integrated to develop automated data analysis tools. We also strongly believe that machine learning could be explored by health researchers to enhance conventional statistics in decision making for added reliable validation measures.

## 1. Introduction

Recently, machine learning has been fueling active discussions among clinicians and health researchers, particularly for decision making in e-diagnosis, disease detection and medical image analysis [1,2,3,4,5]. A few common questions are “can machine learning replace conventional statistics?”, “are they the same?” and “how statistics be integrated with machine learning?”. This review focuses on the concept of conventional statistics and machine learning in health research and the explanation, comparison and examples may answer the aforementioned questions.

It is seen from various research that conventional statistics have dominated health research [6,7,8,9,10,11,12,13,14,15]; however, machine learning, since its inception, is widely being used by data scientists in various fields [16,17,18,19,20,21,22,23,24,25,26,27]. Examples of common conventional statistical analyses are hypothesis testing (t-test, ANOVA), probability distributions (regression) and sample size calculation (hazard ratio), whereas in machine learning, the common concepts are model evaluation, variable importance, decision tree, classification and prediction analysis. While many industries are capturing the vast potential of machine learning, healthcare is still slow in attaining the optimum level to make sense of newer technologies and computational methods. This could be due to uncertain reliability and trust in machines to analyze big data and make timely decisions on patients’ health. The production of large amounts of healthcare data [28,29], such as administrative data, patients’ medical history, clinical data, lifestyle data and other personal data, makes their analyses unmanageable using basic statistical software tools, which subsequently leads to the need of technologically advanced applications with cost-efficient high-end computational power. More importantly, the applications must meet the target or be designed with user-friendly interfaces and data protection utilities to aid the end users, who are the biostatisticians, researchers and clinicians. The experts need automated systems to optimize diagnoses and treatments, enhance prognostications, predict health risks and develop long-term care plans. In line with this, we need to answer the important question of whether machine learning is completely different from conventional statistics or they are correlated with each other. 

To begin with, conventional statistics have a history of over 50 years, beginning in the early 17th and 18th centuries, when mathematical theories were introduced by various scientists. In the 18th century, the importance of advanced statistics in medicine was a prominent topic, where more theories were integrated to invent inferential statistical models. Later, the use of computational power in statistical analysis was given priority, hence advanced software tools were developed. Machine learning was introduced in 1952, and recently it has advanced into deep learning and is used as the basis of artificial intelligence [30,31,32]. Figure 1 describes the evolution of conventional statistics and machine learning in health research.

### 1.1. Past Reviews, Rationale for the Review and Intended Audience

There are recent reviews on the comparison between conventional statistics and machine learning [33,34,35,36,37]. These reviews have presented the definitions of the two terms and the advantages and disadvantages over each other. However, to the best of our knowledge, the conceptual integration between these two fields using examples in health research have not been discussed previously. Moreover, the similarities between these two fields have not been investigated thoroughly. This review may clarify the confusion among clinicians as to whether machine learning can be integrated with conventional statistics in health research. Clarifying the integration between conventional statistics and machine learning may be able to convince health researchers to explore this approach in the future. The intended audience of this review is not only healthcare researchers, but statisticians and data scientists as well.

### 1.2. Review Content

This review contains five sections: (i) concepts in conventional statistics and machine learning, (ii) advantages and disadvantages of conventional statistics and machine learning, (iii) a case study of breast cancer survival analysis using a few techniques comparing conventional statistics and machine learning, (iv) simplified machine learning algorithms and their relationship with conventional statistics and (v) a discussion explaining the integration of conventional statistics with machine learning and the significance of machine learning, derived from fundamental conventional statistics. Section (iii) is explained using a proven breast cancer prediction model [38], which has attracted a broad range of readers both from the medical domain and computer science. The terms conventional statistics (CS) and machine learning (ML) are used throughout the review.

## 2. Survey Methodology

The review was conducted using published works related to: i.history of conventional statistics and machine learning in medicineii.comparison between conventional statistics and machine learningiii.use of machine learning in various fieldsiv.analysis of medical data using conventional statistics andv.use of machine learning and artificial intelligence in medical analysis.

The digital libraries and search engines used to extract the literature are Google Scholar, Web of Science and PubMed. The literature search was followed by selecting relevant literature using inclusion and exclusion criteria as listed below.

### 2.1. Inclusion Criteria

(i)all papers with year of publication between 2015 to 2022(ii)all open access papers that are freely available(iii)the keywords used for the search are conventional statistics, machine learning, medical data, comparison and health research. The entries by using these keywords were from various medical domains, machine learning analyses and statistics in healthcare research, not focusing only on one type of disease.

### 2.2. Exclusion Criteria

(i)all papers not relevant to our topic(ii)all papers that are not freely accessible(iii)all papers with year of publication before 2015

The initial total number of papers is 511. The selection process is explained in Figure 2. The final total number of selected literature is 102.

## 3. Results

### 3.1. Concepts in Conventional Statistics

#### 3.1.1. Hypothesis Testing and Statistical Inference for Classification

Hypothesis testing is the interpretation of results by making assumptions (hypotheses) based on experimental data. The statistical tests (e.g., *t*-test, ANOVA) are used to interpret the results based on measures such as *p*-value (significant difference). Biostatisticians and medical scientists perform statistical analysis using conventional software tools [39,40,41,42,43,44,45,46,47,48,49,50,51,52,53,54,55,56] because healthcare providers’ main objective is to focus on analysis based on hypothesis testing in the context of patient care to check if treatments or drugs yield positive outcomes or how to control certain risk factors for a particular disease. Therefore, they barely explore or pay attention to the use of advanced computer science applications and automated predictive tools such as Predict, CancerMath and Adjuvant. [57]. The basic concepts of hypothesis testing are explained in Table 1.

In conventional statistics, the approach used is the conclusion or “inference” in the form of mathematical equations and measures to make predictions. For instance, an inferential work to assess unknown evidence from observed data could be achieved via a hypothesis-testing framework. The aim of hypothesis testing is to reject the null hypothesis if the evidence found is true and clinically significant. For example, in deciding which surgical treatment, “does breast-conserving therapy or mastectomy promote better survival among breast cancer patients?” is an inferential question and the answer is unobservable. In this scenario, patients are considered the observation, whereas the treatment types and survival data are the independent variables, which decide the inference [34]. The results of the analysis classify the dependent variable (surgical treatment) based on the patterns of independent variables. 

#### 3.1.2. Regression

Regression analysis is a set of statistical methods to estimate the relationship between a dependent variable and a set of independent variables. Regression has been widely used in healthcare research to analyze and make predictions on various diseases. Selection of a particular type of regression depends on the type of dependent variable, such as continuous and categorical. Linear regression is used to determine the relationship between a continuous dependent variable and a set of independent variables. This analysis estimates the model by minimizing the sum of squared errors (SSE). Moreover, nonlinear regression also requires a continuous dependent variable, but this is considered advanced as it uses an iterative algorithm rather than the linear approach of direct matrix equations [58].

A categorical dependent variable is analyzed using logistic regression. This analysis transforms the dependent variables which have values of distinct groups based on specific categories and uses Maximum Likelihood Estimation to estimate the parameters. Logistic regression is further divided into binary, ordinal and nominal categories. A binary variable has only two values, such as survival status (alive or dead), an ordinal variable has at least three values in order, such as cancer stage (stage 1, stage 2, stage 3), whereas a nominal variable has at least three values which are not categorized in any order, such as treatment (chemotherapy, radiotherapy, surgery) [38].

### 3.2. Concepts in Machine Learning (ML)

#### 3.2.1. Predictive Analytics

Prediction works with the concept of modeling using machine learning algorithms. This prediction model requires a reliable relationship between the observations (patients) and variables (independent variables). Prediction models generate accuracy measures to determine the quality of data and predict the final outcome using the observations (patients), input data (independent variables) and output data (dependent variable). The basic concepts of predictive analysis are explained in Table 2.

#### 3.2.2. Representation Learning

Representation learning is the process of training machine learning algorithms to discover representations which are interpretable. Different representations can entangle various explanatory factors in a specific dataset. The outcome of representation learning should ease the subsequent task in the decision-making process. For example, representation learning handles and groups very large amounts of unlabeled training data in unsupervised or semi-supervised learning. The grouping of the unlabeled data is used for the corresponding task, such as feature selection and decision tree, to predict outcomes. The challenging factor of representation learning is that it has to preserve as much information as the input data contains in order to attain accurate predictions. Healthcare research utilizes representation learning mostly in image recognitions, such as biomedical imaging-based predictions [59].

#### 3.2.3. Reinforcement Learning

Reinforcement learning (RL) trains machine learning models to make a sequence of decisions, unlike supervised learning, which relies on a one-shot or single dependent factor. Its main objective is to endow an individual’s skills to make predictions through experience with the environment around them and develop evaluative feedback. This unique feature of reinforcement learning helps in providing prevailing solutions in various healthcare diagnosis and treatment regimens which are usually characterized by a prolonged and sequential procedure. Reinforcement learning follows a few techniques for sequential decision making, namely, efficient techniques such as experience-level, model-level and task-level and representational techniques such as representation for value function, reward function and task or models). Applications of RL in different healthcare domains, such as chronic diseases and critical care, especially sepsis and anesthesia, are explained in detail in [60,61].

#### 3.2.4. Causal Inference/Generative Models

Causal inferencing plays a vital role in understanding the mechanisms of variables to find a generative model and predict outcomes which the variables are subjected to. The characteristic of causal inference is to find answers for questions about the mechanisms by which the variables come to take on values. For example, epidemiologists gather dietary-related data and find the factors affecting life expectancy to predict the effects of guiding people to change their diet. Examples of causal models are models with free parameters (fixed structure and free parameters) and models with fixed parameters (free parameters with values). A large number of variables, small sample size and missing values are considered serious impediments to proper data analysis and production of accurate decision making in the medical domain. Causal inferencing is used in healthcare research mainly for clinical risk prediction and improving accuracy of medical diagnosis, despite the issues with data [62].

### 3.3. Advantages and Disadvantages of Conventional Statistics and Machine Learning

#### 3.3.1. Data Management

Conventional statistics are more suitable for simpler datasets, whereas machine learning can manage complex datasets. In general, conventional statistical analyses are performed if the research has prior literature about the topic of interest, the number of variables involved in the study is relatively small and the number of observations (samples) is bigger than the number of variables. This may assist the scientist in understanding the topic conveniently by selecting the important variables from prior knowledge, as well by applying appropriate analytical models to check the association between variables (independent) and outcomes (dependent). The conventional statistical approach gives more priority to the type of dataset, for example, those including a cohort study, which follow a specific hypothesis [35]. On the other hand, prediction analysis based on machine learning algorithms learn from data without relying on rules-based programming, which does not make any prior assumption, but is rather based on the original data provided. Machine learning algorithms can handle multi-dimensional big data, but conventional statistics can handle only one specific format of data at a time. Furthermore, machine learning algorithms can handle data from different data sources such as external databases or online data repositories.

#### 3.3.2. Computational Power, Interpretation/Explainability and Visualization of Results

The analytical strategy of biostatisticians depends on their pathophysiological knowledge and experience where data-driven prediction analysis challenges this paradigm of thought, and the increasing computational power may unmask the associations not realized by the human mind. 

In conventional statistical analysis, scientists use basic software tools, which lack the capability to handle big data and visualization of results. Machine learning black box algorithms have the ability to uncover subtle hidden patterns in multi-model data [63]. However, interpretability is domain-specific, hence the visualization techniques play a vital role in explaining the results to higher stakeholders. Conventional statistical software tools produce basic visualization, whereas the advanced data analytics tools produce domain-specific, customized, inherently interpretable models and results [63]. Machine learning is often very complex and difficult to be interpreted by clinicians because it uses computational programming and not a user-friendly tool such as SPSS. Conventional statistics are easily interpretable and have lower capacity, thus present a smaller risk of failing to generalize non-causal effects. 

Conventional statistics are considered more computationally efficient and more readily acceptable in the medical domain. Contrarily, the results and visualizations produced by ML algorithms are different from the CS methods, and no proper guideline is available on the ways to explain the graphs for interpretation of final results. Machine learning requires high computational power in terms of processing power and storage. Moreover, ML algorithms are updated regularly into newer versions, which requires updates in coding. Furthermore, ML models have the ability to over-predict (overfitting), where the predicted model is closely related to the provided dataset. This could constrain the possibility to generalize the model in different datasets to produce better accuracy. Therefore, validity is required in both cases to finalize decisions. ML algorithms are able to provide required results and decisions automatically from precise training data based on their built-in functions from the programming tools. Nevertheless, when dealing with large amount of data, more hybrid models can be designed to resolve the issues arising in data science for knowledge extraction, especially in healthcare. There is a myriad of algorithms and software for ML techniques to build prediction models on diseases. In medical informatics, R, Matlab, Waikato Environment for Knowledge Analysis (WEKA) toolkit and Python [64,65,66,67,68,69,70] are a few of the widely used programming languages and software in conducting prediction analysis. In the need of viable decisions and interpretation, healthcare providers and researchers can consider leveraging explainable ML models, instead of focusing only on the results. If the interpretability in a domain-specific notion can be followed by researchers, the level of trust on black boxes among clinicians could be improved.

#### 3.3.3. Dimensionality Reduction

Dimensionality reduction involves reduction of either dimension of the observation vectors (input variables) into smaller representations [71]. Technically, dimensionality reduction transforms original dataset A of dimensionality *N* into a new dataset B of dimensionality *n* [72]. Machine learning models follow various dimensionality reduction techniques based on the types of data in a specific research analysis. The larger the number of input variables, the greater the complication in the predictive models; thus, dimensionality reduction helps to select the best input variables to predict the models. A few methods of reduction are Principal Component Analysis (PCA), Kernel PCA (KPCA), t-distributed Stochastic Neighbor Embedding (t-SNE) and UMAP. Dimensionality reduction techniques remove irrelevant input variables from the dataset, which could increase the accuracy of machine learning models. It also helps to eliminate multi-collinearity, which enhances the way of interpreting the variables. In line with this, the dataset with the relevant input variables saves storage space, and less computing power is needed to analyze the data [73].

#### 3.3.4. Frequently Used Models or Methods for Data Assessment

The most frequently used models for the association study in conventional statistics is logistic regression or Cox regression models for binary outcomes, linear regression for continuous outcomes and more extensive models such as generalized linear models based on the distribution of data. This scenario is typically popular in studies addressing public health significance, especially when the analysis involves a population study [74,75,76,77]. Statisticians believe that, in order to draw a firm conclusion or inference, the number of observations in an association study plays an important role [34]. This is a direct approach in hypothesis testing.

Machine learning models are able to capture high-capacity relationships and they are amenable to more operational tasks rather than direct research questions; thus, more research gaps could be solved through the one-stop analysis [38]. Various medical data analyses used a machine learning approach to make decisions [78,79,80,81,82,83,84,85,86,87,88]. Biostatisticians are in a need of an updated methodology that uses a machine learning approach to conduct analysis on a variety of medical data [89]. In this case, the similar concepts in CS and ML need to be emphasized. Machine learning algorithms serve as alternatives to the conventional statistics for common analyses, such as determining effect size, significant factors, survival analysis and imputations. While conceptually they are similar, they are distinct in terms of methods. The core differences between CS and ML concepts are described in Table 3.

### 3.4. Case Study to Compare Conventional Statistics and Machine Learning

A breast cancer dataset from the University Malaya Medical Centre (UMMC), *n* = 8066, diagnosed between 1993 and 2017, was used to perform prediction analysis using both conventional statistics and machine learning. Written informed consent was obtained from the participants included in this study. This dataset was extracted from the cancer registry within the electronic medical record system of UMMC called iPesakit. A total of 23 independent variables and survival status (dependent variable) were used to determine the most important prognostic factors of breast cancer survival. The data description is provided in Table 3. The methods and results from three different types of analysis are compared. SPSS was used to perform conventional statistics and R was used to perform machine learning. The R codes used for machine learning analysis stated in the case study of this paper are deposited on GitHub [90].

#### 3.4.1. Imputation and Data Pre-Processing

Imputation applies both to conventional statistics and machine learning during data cleaning. Single or multiple imputations can be performed using conventional statistical software and programming tools such as R. In this case study, imputation was performed on the dataset to fill the missing values only for conventional statistical analysis. This is because the machine learning algorithms are able to handle the data with missing values. The dataset was split into testing (30%) and training (70%) for machine learning.

#### 3.4.2. Significant Factors (CS) and Variable Importance (ML)

The objective of this analysis was to compare conventional statistics and machine learning (variable importance) to determine the similarities and differences in the results using the same dataset. Table 4 shows the results using significant factor analysis in SPSS. The results from the chi squared test (categorical variable) and Mann–Whitney U test (continuous variables) show that all the independent variables are statistically significant (*p*-value < 0.05).

Figure 3 shows the variable importance plot using random forest VSURF and randomForestExplainer packages in R [72]. The variables are ranked based on variable importance mean from highest to lowest. A threshold was set up to 0.01 and six variables were selected as the most important prognostic factors of breast cancer survival.

The difference between significant testing and variable importance is that the order of the importance is determined in variable importance, but only the status of significance (statistically significant or not) could be determined using the significant factor analysis.

#### 3.4.3. Survival Analysis

Survival analysis in machine learning follows exactly the same concept as the conventional statistics, which is the Kaplan–Meier (KM) estimator. The time series data, date of diagnosis, date of death and date of last follow-up are used to calculate the overall survival rate. The methods used are different; in machine learning, the KM estimator is encapsulated into a single package called survival in R. Programming codes are used to plot the survival curve directly by specifying the variables. In contrast to conventional statistics, it is not an algorithm, but a type of data analysis where the time series data are selected to plot survival curves with a life table and hazard ratio. Both conventional statistics and machine learning follow the same rules to predict survival rate. The survival curves are shown in Table 5. Survival curves are created for three variables: tumor size, cancer stage and positive lymph nodes. The survival curves from SPSS and R produced quite similar results in terms of survival rate for various categories in each variable, but with differences in numerical values (survival percentages).

### 3.5. Simplified Machine Learning Algorithms and Their Relationship with Conventional Statistics 

The mathematical equations in conventional statistics are encapsulated to form algorithms in machine learning. These algorithms are used to perform predictions using supervised and unsupervised machine learning. The integration between the mathematics behind conventional statistics and machine learning are explained using the techniques, model evaluation (supervised learning), variable importance (supervised learning) and hierarchical clustering (unsupervised learning). A proven breast cancer prediction model [38] has been used to explain the concepts of the algorithms. 

Model evaluation in machine learning is similar to power analysis in conventional statistics for assessing the quality of data. It is the key step in machine learning, as the ability of the model to make predictions on unseen or future samples enhances the trust on the model to be used in a particular dataset. The measurement for model evaluation is the accuracy in percentage (estimate of generalization of a model on prospective data). Six different supervised machine learning algorithms (decision tree, random forest, extreme gradient boosting, logistic regression, support vector machine, artificial neural networks) are simplified. These algorithms have been widely used in medical informatics [65,67,91,92,93].

#### 3.5.1. Decision Tree

Decision tree has been widely used in medical informatics [63,71], as it is the basic concept used by other algorithms such as random forest and gradient boosting, but with certain differences in the processes to predict the final output. 

The decision tree algorithm follows the model of a tree structure, where it has a root node, decision node and terminal node. The root node starts with the most important independent variable followed by decision nodes (other independent variables). The terminal node indicates the dependent variable, which is the final predicted output. 

The processes in the decision tree are summarized into three steps: (i) choosing features, (ii) setting conditions to split and (iii) stopping the splitting process to produce a final output. A tree structure is built based on the observation falling in each region and the mean value of prediction. 

The splitting process is continued until a user-defined stopping criteria (the number of observations per node) is reached. In the case of more than two variables, the regions cover all the variables with multiple axes.

#### 3.5.2. Random Forest

Random forest is an ensemble learning algorithm, which is derived from decision tree. It follows the rule of decision tree, but constructs a multitude of decision trees at training time and outputs the class with the maximum vote. Random forest is the state-of-the-art algorithm in medical informatics, as it has the ability to manage multivariate data [72].

Random forest is known as an improved version of decision tree, as it constructs more than one tree to select the best output, whereas decision tree constructs only one tree. The number of trees constructed during the training process is not default, as the users can specify it based on the number of samples. The number of trees is directly proportional to the number of samples.

#### 3.5.3. Extreme Gradient Boosting

Gradient boosting follows the principle of random forest but with an added interpretation to predict the final output. This algorithm also constructs multiple trees called boosted trees. A prediction score is assigned for each leaf in the boosted trees (gradients), whereas random forest only contains the final decision value for one tree. Several studies have used the gradient boosting approach to analyze medical data [73,92].

This algorithm also considers the weak and strong prediction values during training before making the final decision, unlike decision tree and random forest, which only select the tree with the best class, without considering the other classes. This method in gradient boosting is known as the impurity measure. The scores of all the leaves in the trees are summed up to produce the gradient values, and the final prediction is made based on the mean value, called gradient boosting.

#### 3.5.4. Logistic Regression

Most studies use regression for prediction analytics in medicine [61]. Logistic regression predicts categorical output, for example, the survival status (alive or dead). The predictions are made based on the probabilities shown by a curve. This process is repeated for all the samples. The curve is shifted to calculate new likelihoods of the samples falling on that line. Finally, the likelihood of the data is calculated by multiplying all the likelihoods together and the maximum likelihood is selected as the final result.

#### 3.5.5. Support Vector Machine

Just like all other algorithms, support vector machine (SVM) segregates data into different classes, but it involves discovery of hyperplanes. The hyperplane divides the data into two groups (classes). The points closer to the decision boundary or hyperplane are called support vectors. The final prediction is made based on the values of independent variables and the support vectors corresponding to the hyperplane. The number of hyperplanes depends on the number of independent variables. The SVM structure is complicated, with more than three features, but its ability to process multiple variables with multiple hyperplanes at a time to predict the final outcome is one of the advantages of this algorithm.

#### 3.5.6. Artificial Neural Networks

Neural networks are an artificial representation of the human nervous system. It can be explained using the structure of neurons and how they work. The dendrites collect information from other neurons in the form of electrical impulses (input). The cell body generates inferences based on the inputs and decides the actions to be taken. The outputs are transmitted through exon terminals as electrical impulses to other neurons.

The same concept is implied in artificial neural networks (ANN). The inputs refer to the independent variables and samples provided to the algorithm. The inputs are multiplied by weights to calculate the summation function. The higher the weight an input has, the more significant the input is to predict the final output. The activation function predicts the probabilities from the training data and generates a final outcome. This is known as a single-layer perceptron. There are three types of layers in ANN, which are input layer, hidden layer and output layer. 

Model evaluation is followed by variable importance in machine learning. Variable importance (importance score) is an alternative to identifying the significant factors (*p*-value) in conventional statistics using confidence interval measure and hypothesis testing. After performing model evaluation, the elements (variables) of the input data need to be explored further in regard to how they contribute to the accuracy measure. Hence, machine learning algorithms are built-in with a technique called variable importance or feature importance to analyze the variables or features in the input data. The distribution of these variables contributes to the prediction of the final outcome using machine learning models. 

## 4. Discussion

### 4.1. Integration of Conventional Statistics with Machine Learning

Statistics is a branch of mathematics that consists of a combination of mathematical techniques to analyze and visualize data 90. On the other hand, machine learning is a branch of artificial intelligence that is composed of algorithms performing supervised and unsupervised learning. The comparison or integration between conventional statistics and machine learning has gained momentum over the last few years [94,95]. It is plausible that data integrity with protection is the most challenging task in healthcare analytics [96,97,98]. Hence, from this review, it is found that the integration between these two fields could unlock and outline the key challenges in healthcare research, especially in handling the valuable asset called data. Individuals should not be subject to a final decision based solely on automated processing or machine learning using algorithms, but integration of statistics and human decision making is essential at an equal rate. The integration between statistics and machine learning is shown in Figure 4. 

### 4.2. Significance of Machine Learning to Healthcare, Education and Society

The review on the integration between conventional statistics and machine learning is the key factor to convince clinicians and researchers that machine learning algorithms are based on core conventional statistical ideas; thus, they could be used to supplement data analysis using conventional statistics. From this review, we believe that machine learning, which follows the fundamentals of conventional statistics, has a positive impact on healthcare. The significance of machine learning to healthcare is explained (Figure 5).

Prior to the emergence of the data deluge, healthcare providers made clinical decisions based on formal education and their experience over time in practice. Decision analysis in healthcare has been criticized because the experience and knowledge of the decision makers (clinicians) on patient characteristics are not the same or standardized. The linear process of the decision-making model involves four steps, which are data gathering, hypothesis generation, data interpretation and hypothesis evaluation. All four steps require data from different departments, clinicians from different expertise and various data analytical methods to make the final decision. Experienced clinicians may not deliberately go through each step of the process and may use intuition to make decisions, instead of facing obstacles handling several hypotheses with different personnel. As this is applied to experienced clinicians, a novice clinician would have to understand and rely on the analytical principles and theory behind a decision analysis process in a particular situation handling a patient. In this case, the healthcare sector is in need of clinical decision support tools to enhance and standardize clinical decision making. 

Machine learning algorithms are widely used to develop clinical decision support tools. These algorithms compile the four steps (data gathering, hypothesis generation, data interpretation and hypothesis evaluation) of traditional decision making into one. The advantages of machine learning algorithms in medical informatics depend on the objectives of the research and the types of data used. ML algorithms such as decision tree, random forest, gradient boosting, regression, support vector machine and artificial neural networks are suitable for medical informatics, as they are able to handle big data, a combination of numerical and categorical data and missing values. Moreover, these algorithms generate visualizations, which could be transformed automatically (integrated into tools) to be used by the clinicians as guidelines for patients. 

In any machine learning analysis, domain experts are still required to enhance the reliability of the machine and make sense of the results. In medical informatics specifically, the decision of clinicians on a particular patient’s health condition plays an important role in giving suggestions to the patient. The automated decision support tools may help clinicians in decision making to save time and costs, and to follow a standard procedure to prevent conflict in final decisions. 

The field of medicine relies heavily on knowledge discovery and understanding of diseases associated with the growth in information (data). Diagnosis, prognosis and drug development are the challenging key principles in medicine, especially in complex diseases, such as cancer [91]. Based on the principal of evidence-based medicine, decision making based on data and validation should be more agile and flexible to better translate the basic knowledge of complexities into growing advances. The integration of conventional statistics and machine learning to clinical applications should be carefully adopted with a collaborative efforts that includes all major stakeholders for the positive influence of machine learning in medicine [91].

Comparison between the basic workflow of conventional statistics and machine learning is explained in Figure 6.

### 4.3. Automation of Machine Learning in Healthcare Research 

The machine learning approach could be transformed into an updated guideline for academicians and researchers. The medical academic sector may use the methodologies for teaching and learning programs to educate medical students on the importance of machine learning. Moreover, researchers in the same field can follow the techniques and machine learning models to conduct research and cohort studies in any healthcare domain. 

Biostatisticians may consider using machine learning techniques and automated tools [92,93] together with conventional statistics in order to improve the performance of analytics and reliability of results. The integration between statistics and machine learning may assist biostatisticians to provide novel research outcomes. 

The automation of machine learning in healthcare analysis has been applied in a recent study by our research group [99]. The automated tools may assist biostatisticians to provide novel research outcomes. 

A guideline to transform statistics and machine learning (derived from the fundamental mathematics of conventional statistics) into an automated decision support tool is illustrated in Figure 7.

This guideline could be a standardized pipeline for the data science community to analyze medical data and to develop artificial intelligence-enabled decision support tools for clinicians and researchers. 

In a decision support tool, (i) the data gathering is replaced by the automated data capture from electronic medical records (EMR) or databases from multiple heterogeneous sources, concomitantly; (ii) hypothesis generation is the specification of input variables (independent and target) and the final outcome based on the research question or a question for clinical decision (output); (iii) data interpretation is done using algorithms such as random forest, support vector machine and neural networks, which have their specific formulas to read the data, clean the data, capture the required variables, analyze the data based on the specified requirements and perform comparative analytics automatically using different algorithms; (iv) finally, hypothesis evaluation is done by producing interactive charts to visualize the final outcomes to make decisions efficiently. All these steps are performed in a streamlined environment often referred to as automated clinical decision making, which saves the effort of engaging different experts and analytical platforms. The experience which clinicians traditionally use to make decisions is replaced by the legacy data the algorithms leverage to make decisions.

Most important of all, the data management or completeness of data for automated decision making plays an essential role when it comes to statistics and machine learning. The integration between statistics and machine learning can be used to train automated models with imputed missing values in the data, which would improve the generalizability and robustness of models [100].

The integration between statistics and machine learning does not only contribute to data augmentation, but also to medical diagnostics using multi-model data. In the future, this approach together with deep learning methods is suggested to be used in bioinformatics analysis using genomic data or a combination of genomic and clinical data to enhance the automated decision-making process. Deep learning, being one of the unprecedented technical advances in healthcare research, assists clinicians in understanding the role of artificial intelligence in clinical decision making. Hence, deep learning could serve as a vehicle for the translation of modern biomedical data, including electronic health records, imaging, omics, sensor data and text, which are complex, heterogeneous, poorly annotated and generally unstructured, to bridge clinical research and human interpretability [101,102].

## 5. Conclusions

Conventional statistics are the fundamentals of machine learning, as the mathematical concepts are encapsulated into simplified algorithms executed using computer programming to make decisions. Machine learning has the added benefit of automated analysis, which can be translated into decision support tools, providing user-friendly interfaces based on interactive visualizations and customization of data values. Such tools could assist clinicians in looking at data in different perspectives, which could help them make better decisions. Despite the debate between conventional statistics and machine learning, the integration between the two accelerates decision-making time, provides automated decision making and enhances explainability. This review suggests that clinicians could consider integrating machine learning with conventional statistics for added benefits. Both machine learning and conventional statistics are best integrated to build powerful automated decision-making tools, not limited to clinical data, but also for bioinformatics analyses.

## Figures and Tables

**Figure 1 diagnostics-12-02526-f001:**
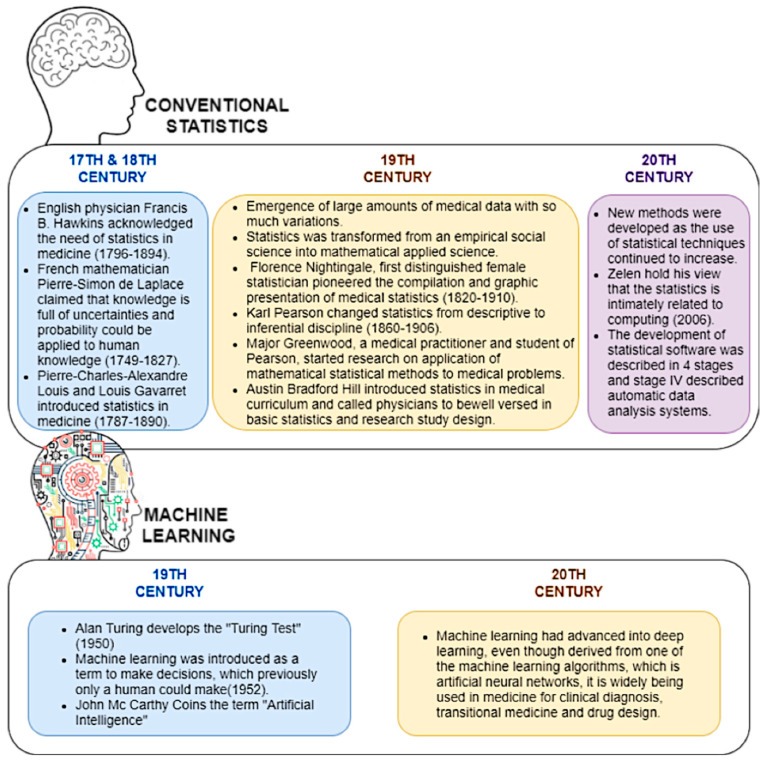
The evolution of conventional statistics and machine learning in health research.

**Figure 2 diagnostics-12-02526-f002:**
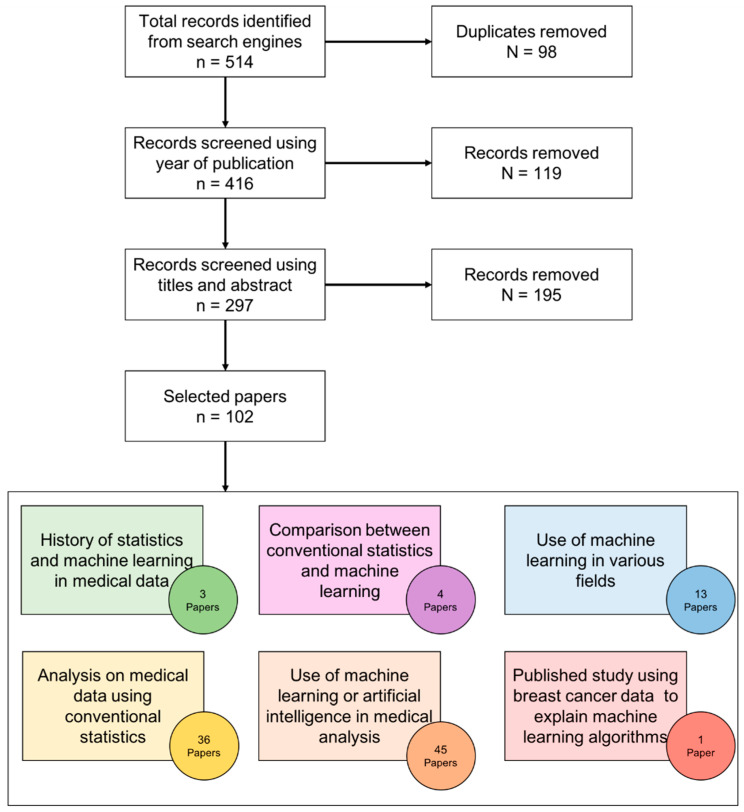
Survey methodology for literature selection.

**Figure 3 diagnostics-12-02526-f003:**
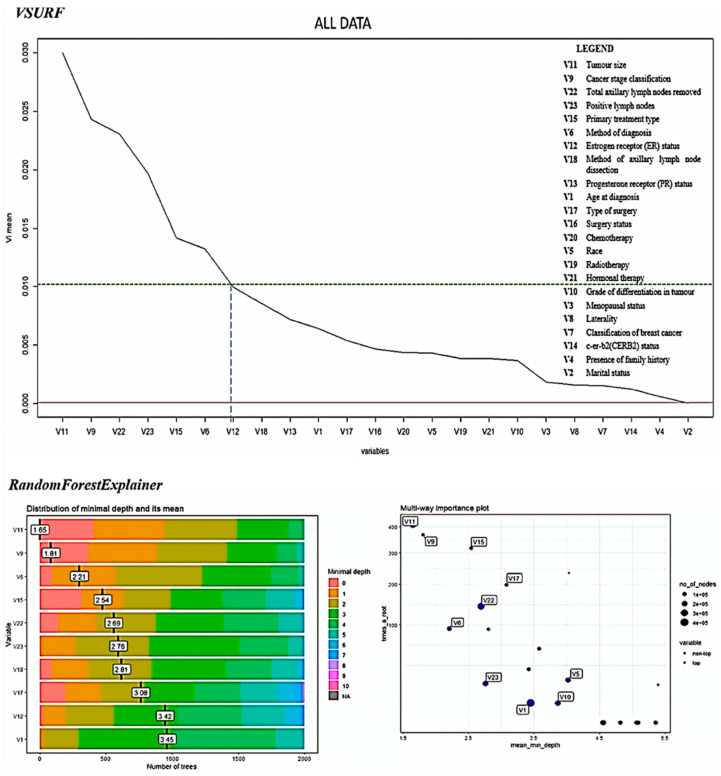
Results of variable importance using VSURF and randomForestExplainer R packages to determine the important factors affecting breast cancer survival.

**Figure 4 diagnostics-12-02526-f004:**
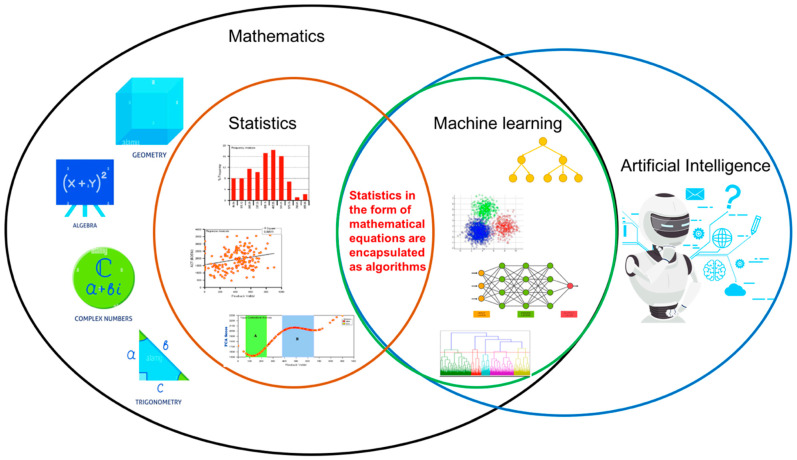
Integration between conventional statistics and machine learning.

**Figure 5 diagnostics-12-02526-f005:**
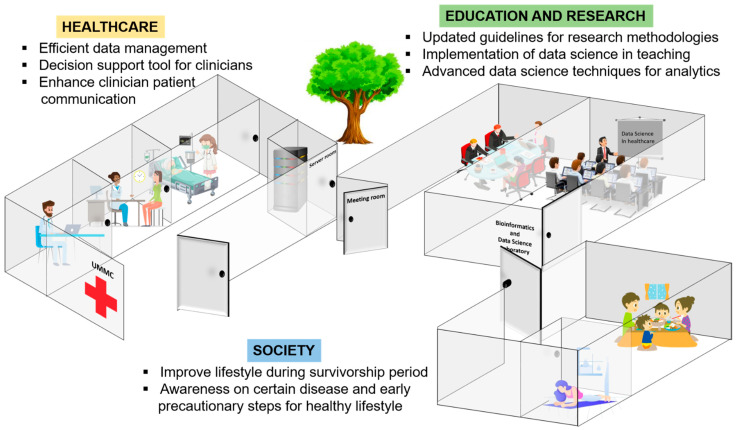
Significance of machine learning to healthcare, education and research and society.

**Figure 6 diagnostics-12-02526-f006:**
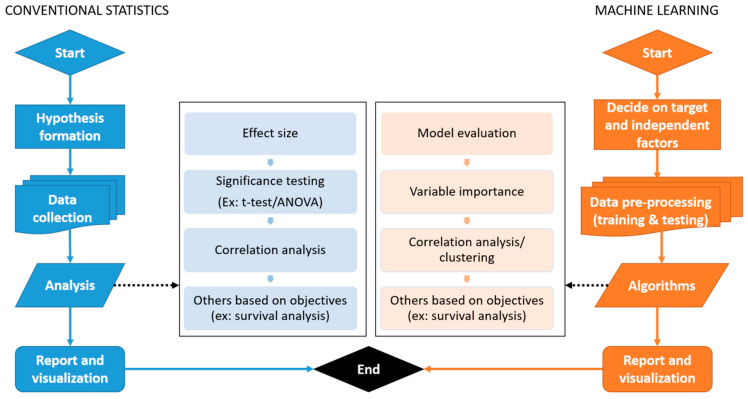
General workflow of conventional statistics and machine learning.

**Figure 7 diagnostics-12-02526-f007:**
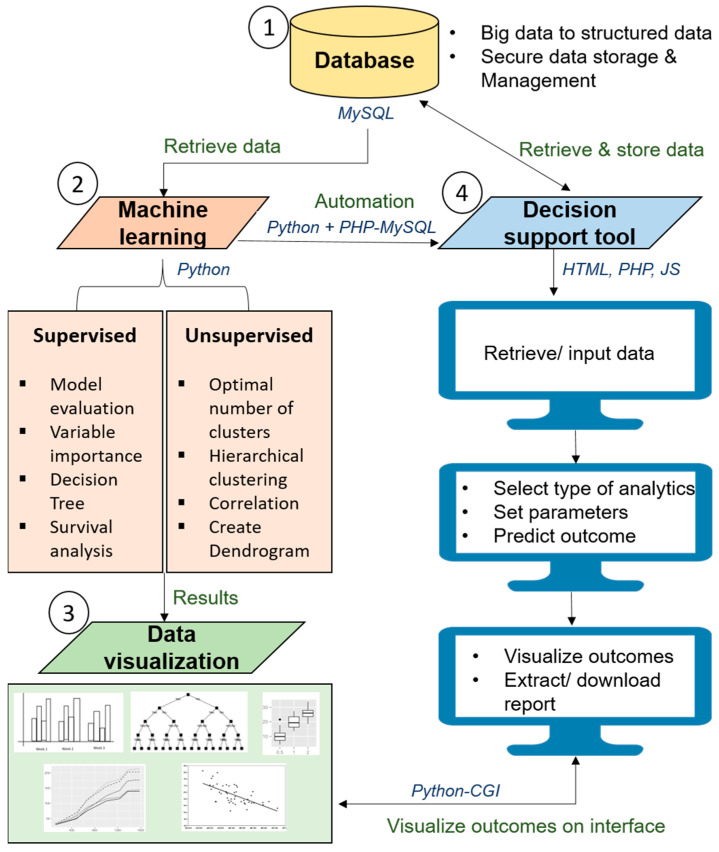
Pipeline to develop automated health research platform with machine learning.

**Table 1 diagnostics-12-02526-t001:** Concepts in hypothesis testing.

Approach	Concept	Procedure
Hypothesis testing • Inference	Research question: • Is the null hypothesis false? • E.g., There is no difference in survival outcome between patients who underwent surgical treatment of mastectomy or breast-conserving therapy Answer: • The null hypothesis is false • E.g., There is a significant difference between type of surgical treatment and survival outcome Decision rule: • A statistical test analyzes the data, which results in a *p*-value, which is then compared against the significance level and probability odds ratio or hazard ratio with a magnitude of confidence interval (CI) • E.g., *p* < 0.05, Hazard ratio 1.50, CI 1.12–2.30 Decision: • Reject the null hypothesis	Step 1: Identify predictors from related literature Step 2: Design a hypothesis and compare the similarities and differences using a new dataset

**Table 2 diagnostics-12-02526-t002:** Concepts in predictive analysis (classification).

Approach	Concept	Procedure
Classification	Research question: Is label 1 considered as the target outcome? Answer: Yes, label 1 is the target outcome Decision rule: • A trained classifier that analyzes an unlabeled observations’ variables and values, which results in a predicted label (1).	Step 1: Split dataset into training and testing datasets Step 2: Train the data using a specific algorithm Step 3: Test the remaining dataset using the trained algorithms to predict the results accurately

**Table 3 diagnostics-12-02526-t003:** Machine learning alternatives to conventional statistics.

Analysis	Conventional Statistics	Machine Learning
**1**	**Imputations**	**Imputations**
Objective	To impute missing value based on pattern of missing data. For example, missing by random	To impute missing values to maintain the quality of data
Method	Missing values are identified on the percentage of missing data, with an acceptance range between (10–20%)	1. Determine the missing values in the data 2. Perform multiple imputations using algorithms such as *mice*, *Amelia* and *missForest* (sample built in packages in R)
Result	Imputed data/clean data	Imputed data/clean data
**2**	**Effect Size**	**Model Evaluation**
Objective	To determine if the data explain the variability in data. Often called residual error, the residual should be as minimized as possible	To determine the quality of data to be used for further analysis
Method	Residual analysis, if necessary, standardized residual error is performed using linear regression	1. Split data into training and testing sets 2 Build models using algorithms (e.g., decision tree, support vector machine, etc.)
Results	Measurable R^2^ • Under-fit model < 0.3 • Good-fit model (0.3–0.7) • Over-fit model (preferable) (> 0.7)	Accuracy, sensitivity, specificity, precision, Matthew Correlation Coefficient, area under the receiver operating curve (AUROC) • Good-fit model (> 0.7)
**3**	**Significant Factors**	**Variable Importance**
Objective	To select important independent variables, which affect the target variable (dependent variable)	To select important independent variables, which affect the target variable (dependent variable)
Method	1 Run the analysis using all data 2 Treat missing values or exclude missing values 3 Chi square test or logistic regression to choose significant variable	1. Run variable importance using the best model, which fit the data from model evaluation 2. Rank and select important variables for further analysis using the importance score
Results	*p*-value, 95%CI OR 2.00 CI (1.51–12.52)	Variable importance score/mean (numerical) and variable importance plot (visualization)
**4**	**Survival Analysis**	**Survival Analysis**
Objective	To determine survival rate using time series data	To determine survival rate (%) using time series data
Method	Similar to machine learning, just the software is different	1. Specify independent variable and target (survival years) 2. Define survival status, event = dead/1 3. Use machine learning survival package, which computes survival based on Kaplan–Meier to calculate survival percentage
Results	Survival rate in percentage, Hazard ratio	Survival rate in percentage, Hazard ratio

**Table 4 diagnostics-12-02526-t004:** Results from significant factor analysis in SPSS.

Variables (Independent)	Total, *n* (%)	Survival Status (Dependent)		*p*-Value ^1^
		Alive, *n* (%)	Death, *n* (%)	
Age (years), median	51 (42, 61)	51 (43, 60)	53 (42, 63)	0.001
Marital status				0.001
Married	6397 (81.6)	4554 (82.5)	1843 (79.3)	
Not married	1443 (18.4)	963 (17.5)	480 (20.7)	
Menopausal status				0.000
Natural menopause	3984 (50.8)	2675 (48.5)	1309 (56.3)	
Pre-menopause	3347 (42.7)	2459 (44.6)	888 (38.2)	
Surgical menopause	509 (6.5)	383 (6.9)	126 (5.4)	
Presence of family history				0.000
No	6357 (81.1)	4378 (79.4)	1979 (85.2)	
Yes	1483 (18.9)	1139 (20.6)	344 (14.8)	
Race				0.000
Chinese	5394 (68.8)	4041 (73.2)	1353 (58.2)	
Indian	921 (11.7)	608 (11.0)	313 (13.5)	
Malay	1525 (19.5)	868 (15.7)	657 (28.3)	
Method of diagnosis				0.000
Excision	1617 (20.6)	1294 (23.5)	323 (13.9)	
FNAC	1886 (24.1)	1013 (18.4)	873 (37.6)	
Imaging only	35 (0.4)	31 (0.6)	4 (0.2)	
Trucut	4302 (54.9)	3179 (57.6)	1123 (48.3)	
Classification of breast cancer				0.000
Insitu	366 (4.7)	348 (6.3)	18 (0.8)	
Invasive	7474 (95.3)	5169 (93.7)	2305 (99.2)	
Laterality				0.000
Bilateral	97 (1.2)	26 (0.5)	71 (3.1)	
Left	3553 (45.3)	2464 (44.7)	1089 (46.9)	
Right	3895 (49.7)	2830 (51.3)	1065 (45.8)	
Unilateral side unknown	295 (3.8)	197 (3.6)	98 (4.2)	
Cancer stage				0.000
Pre-cancer	365 (4.7)	347 (6.3)	18 (0.8)	
Curable cancer	6624 (84.5)	4956 (89.8)	1668 (71.8)	
Metastatic	851 (10.9)	214 (3.9)	637 (27.4)	
Tumour size (cm), median	2.7 (1.6, 4.5)	2.3 (1.5, 3.5)	4.00 (2.5, 8.0)	0.000
Total axillary lymph nodes removed, median	11 (4, 16)	12 (6, 17)	9 (0,16)	0.000
Number of positive lymph nodes, median	0 (0, 2)	0 (0, 1)	0 (0,4)	0.000
Grade of differentiation in tumour				0.000
Good	2548 (32.5)	1631 (29.6)	917 (39.5)	
Moderate	2936 (37.4)	2259 (40.9)	677 (29.1)	
Poor	2356 (30.1)	1627 (29.5)	729 (31.4)	
Estrogen status				0.000
Negative	3198 (40.8)	1936 (35.1)	1262 (54.3)	
Positive	4642 (59.2)	3581 (64.9)	1061 (45.7)	
Progesterone status				0.000
Negative	4157 (53.0)	2603 (47.2)	1554 (66.9)	
Positive	3683 (47.0)	2914 (52.8)	769 (33.1)	
c-er-b2 status				0.000
Positive	1862 (23.8)	1245 (22.6)	617 (26.6)	
Negative	5148 (65.7)	3666 (66.4)	1482 (63.8)	
Equivocal	830 (10.6)	606 (11.0)	224 (9.6)	
Primary treatment type				0.000
Chemotherapy	976 (12.4)	438 (7.9)	538 (23.3)	
Hormone therapy	270 (3.4)	100 (1.8)	170 (7.3)	
Surgery	6140 (78.3)	4812 (87.2)	1328 (57.2)	
None	454 (5.8)	167 (3.0)	287 (12.4)	
Surgery status				0.000
Surgery done	6740 (86.0)	5121 (92.8)	1619 (69.7)	
No surgery	1100 (14.0)	396 (7.2)	704 (30.3)	
Type of surgery				0.000
Breast-conserving surgery	1916 (24.4)	1661 (30.1)	255 (11.0)	
Mastectomy	4821 (61.5)	3456 (62.6)	1365 (58.8)	
No surgery	1103 (14.1)	400 (7.3)	703 (30.3)	
Method of axillary lymph node dissection				0.000
Yes	5553 (70.8)	4048 (73.4)	1505 (64.8)	
SLNB (sentinel lymph node biopsy)	540 (6.9)	531 (9.6)	9 (0.4)	

^1^*p*-value reported based on chi square test for all categorical variables and Mann–Whitney U test for continuous variables.

**Table 5 diagnostics-12-02526-t005:** Survival analysis comparison using SPSS and R.

Comparison	SPSS						R
Method	1. Under survival, life table is used to plot the survival curve 2. Log rank test is used to determine significant difference between variables 3. Stage is grouped as Stage 0 (pre-cancer), Stages 1–3 (curable cancer), Stage 4 (metastatic cancer) 4. The grouping of tumor size and positive lymph nodes were done using clinical guideline						1. Package *survival* is loaded 2. Survival years and survival status are used to calculate overall survival rate for selected variables 3. Stage is grouped as Stage 0 (pre-cancer), Stages 1–3 (curable cancer), Stage 4 (metastatic cancer) 4. The grouping of tumor size and positive lymph nodes were done using results from decision tree
Results	a. Tumor size						a. Tumor size
	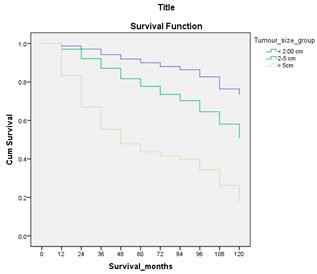						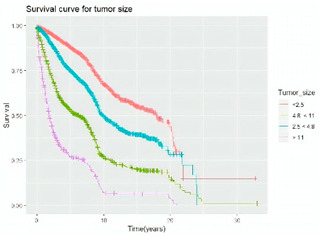
	b. Cancer stage						b. Cancer stage
	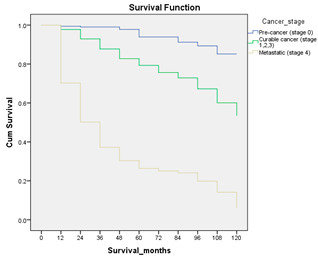						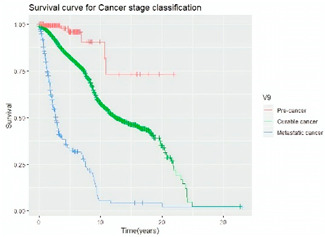
	c. Positive lymph nodes						c. Positive lymph nodes
	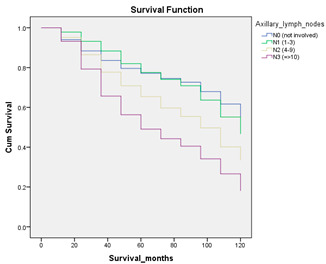						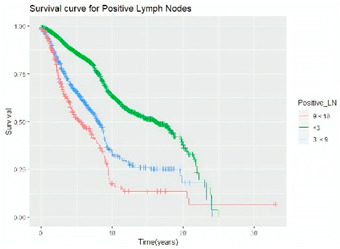
Interpretation	*p*-value < 0.05 means it is statistically significant						The survival percentages are extracted from the survival curve to estimate the survival rate of patients.
		Log Rank (Mantel-Cox)	Chi-square	df	Sig.		
		Tumor size	1105.407	2	0.000		
		Cancer stage	1721.517	2	0.000		
		Positive lymph nodes	234.999	3	0.000		

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
