# Peer review of "Theory and Practice of Integrating Machine Learning and Conventional Statistics in Medical Data Analysis"

_diagnostics, 2022, doi:10.3390/diagnostics12102526_

Round 1

Reviewer 1 Report

The manuscript is well-organized and well-written. However, the reviewer recommend that following two issues are improved.

1. The main topic of the manuscript is medical data analysis, but the description related to machine learning is too general. Please more specify and explain with many more examples based on medical data analysis.

2. Some text within figures is too small. Please clarify the figures.

Author Response

The revision details are attached. 

Reviewer 2 Report

This study reviews the conventional statistics and machine learning methods. the manuscript was well written. However, they have not discuss on deep learning methods which are the most important topic in AI and they are being became better diagnostic tool in bioinformatics. In addition all DL methods use many conventional statistics methods during their processes. 

comparison between conventional statistics 24 and machine learning methods

- Section 2: The search strategy with keywords should be mentioned in detail. In addition, what databases they used for the search?

 - Dimensionality reduction: this is not a disadvantage of statistical methods as they only analyze one feature at a time. 

Author Response

The revision details are attached. 

Round 2

Reviewer 2 Report

Accepted